# MS2Lipid: A Lipid Subclass Prediction Program Using Machine Learning and Curated Tandem Mass Spectral Data

**DOI:** 10.3390/metabo14110602

**Published:** 2024-11-07

**Authors:** Nami Sakamoto, Takaki Oka, Yuki Matsuzawa, Kozo Nishida, Jayashankar Jayaprakash, Aya Hori, Makoto Arita, Hiroshi Tsugawa

**Affiliations:** 1Department of Biotechnology and Life Science, Tokyo University of Agriculture and Technology, 2-24-16 Naka-cho, Koganei-shi, Tokyo 184-8588, Japan; nsakamoto@st.go.tuat.ac.jp (N.S.); s237226y@st.go.tuat.ac.jp (T.O.); fr3701@go.tuat.ac.jp (Y.M.); knishida@go.tuat.ac.jp (K.N.); 2Graduate School of Global Food Resources, Hokkaido University, Kita-9, Nishi-9, Kita-ku, Sapporo 060-0809, Japan; jayashankar.jayaprakash.q1@elms.hokudai.ac.jp; 3Laboratory for Metabolomics, RIKEN Center for Integrative Medical Sciences, 1-7-22 Suehiro-cho, Tsurumi-ku, Yokohama 230-0045, Kanagawa, Japan; aya.hori@riken.jp; 4Division of Physiological Chemistry and Metabolism, Graduate School of Pharmaceutical Sciences, Keio University, 1-5-30 Shibakoen, Minato-ku, Tokyo 105-8512, Japan; 5Molecular and Cellular Epigenetics Laboratory, Graduate School of Medical Life Science, Yokohama City University, Tsurumi-ku, Yokohama 230-0045, Kanagawa, Japan; 6Human Biology-Microbiome-Quantum Research Center (WPI-Bio2Q), Keio University, 35 Shinanomachi, Tokyo 160-8512, Japan

**Keywords:** untargeted lipidomics, tandem mass spectrum, machine learning, lipid class prediction, microbiota-dependent lipids, human fecal samples

## Abstract

**Background**: Untargeted lipidomics using collision-induced dissociation-based tandem mass spectrometry (CID-MS/MS) is essential for biological and clinical applications. However, annotation confidence still relies on manual curation by analytical chemists, despite the development of various software tools for automatic spectral processing based on rule-based fragment annotations. **Methods**: In this study, we present a novel machine learning model, MS2Lipid, for the prediction of known lipid subclasses from MS/MS queries, providing an orthogonal approach to existing lipidomics software programs in determining the lipid subclass of ion features. We designed a new descriptor, MCH (mode of carbon and hydrogen), to increase the specificity of lipid subclass prediction in nominal mass resolution MS data. **Results**: The model, trained with 6760 and 6862 manually curated MS/MS spectra for the positive and negative ion modes, respectively, classified queries into one or several of 97 lipid subclasses, achieving an accuracy of 97.4% in the test set. The program was further validated using various datasets from different instruments and curators, with the average accuracy exceeding 87.2%. Using an integrated approach with molecular spectral networking, we demonstrated the utility of MS2Lipid by annotating microbiota-derived esterified bile acids, whose abundance was significantly increased in fecal samples of obese patients in a human cohort study. This suggests that the machine learning model provides an independent criterion for lipid subclass classification, enhancing the annotation of lipid metabolites within known lipid classes. **Conclusions**: MS2Lipid is a highly accurate machine learning model that enhances lipid subclass annotation from MS/MS data and provides an independent criterion.

## 1. Introduction

Lipids play vital roles in cellular membranes, signaling molecules, and energy storage in living organisms. Dysfunction in lipid metabolism is associated with human diseases [1]. The ClassyFire chemical classification program [2] categorizes over 48,000 molecules as “lipids and lipid-like molecules”, as recorded in LIPID MAPS [3] (as of 6 January 2024). Mass spectrometry-based lipidomics techniques are currently used to elucidate the diversity and abundance profile of lipids. Untargeted lipidomics, utilizing liquid chromatography coupled with tandem mass spectrometry (LC-MS/MS) and electrospray ionization (ESI), efficiently provides the profile of 500 to 1000 lipid molecules from a single biospecimen [4]. The structure of these lipids is elucidated by the pattern of product ion spectra generated by collision-induced dissociation (CID) [5].

Numerous software programs support spectral mining for lipid structure elucidation [6]. Rule-based algorithms that confirm the presence of diagnostic ions to characterize lipid main classes, subclasses, and acyl chain compositions are commonly recommended for lipid structure description, guided by lipidomics standards initiative [7]. The term “lipid subclass” refers to a chemical class defined by the acyl chain linkage (e.g., ester, ether, and vinyl ether) in addition to the main class definition, such as phosphatidylcholine (PC) and phosphatidylethanolamine (PE). LIPID MAPS contains over 300 lipid subclass terms [3]. However, the current diagnostic tools are inadequate for determining lipid subclasses, as misannotations can occur due to contaminant ions from co-eluted lipids, in-source fragments, cluster ions, and spike noises generated by MS detectors [8]. Therefore, an orthogonal criterion is needed to improve annotation confidence in untargeted lipidomics.

Since the launch of the MS-DIAL 4 environment in 2020 [9], the program has been used in our laboratory for 82 projects involving 16,614 biological samples. Lipid annotations have been performed using a decision tree-based fragment annotation program for 117 lipid subclasses with a 1.5 min retention time (RT) tolerance from reference RT values predicted by a machine learning method, as described in the previous report [8]. The MS-DIAL program has provided lipid names for a total of 82,397 MS/MS spectral data in the alignment tables. The annotation results have been manually curated, with 39,871 spectral records labeled as “confidence” and 42,526 spectra labeled as a “mixture of several lipids” or “misannotation (false hit)”.

In this study, we developed a machine learning model called MS2Lipid using correctly labeled spectral records to predict lipid subclasses. Machine learning research on mass spectrometry data is actively conducted in metabolomics (including lipidomics) and proteomics fields [10]. In proteomics, there is a strong effort to construct machine learning models that use spectrum data obtained from biological samples to improve spectrum generation models and annotation accuracy [11]. However, in the field of metabolomics, most studies still primarily utilize spectra of authentic standards as training data. Our study is the first to employ manually curated lipid spectra derived from biological samples as training data in lipidomics. To evaluate the accuracy and precision of MS2Lipid, we used the CANOPUS program, which utilizes spectra records of authentic standard compounds, as a benchmark [12]. The MS2Lipid program was also evaluated on spectra containing at least two different lipids. Since the training dataset spectra were obtained from a single machine, we evaluated the scalability of MS2Lipid by using spectra from different machines and curators. Finally, we investigated the use of MS2Lipid with a molecular spectral networking approach for annotating previously unknown lipid metabolites in the reanalysis of a public human cohort microbiome study data [13].

## 2. Materials and Methods

### 2.1. Mass Spectral Data for Creating MS2lipid Machine Learning Model

A total of 16,614 samples from 82 projects were analyzed using the same analytical method [14] and processed in MS-DIAL 4, where 117 lipid subclasses could be characterized with the rule-based annotation program in combination with less than 1.5 min retention time tolerance matched with predicted retention times of lipids. All experimental data were acquired using reverse-phase liquid chromatography coupled with a SCIEX TripleTOF 6600 system, according to a previously described protocol [13]. Lipid structures were described using the MS/MS spectra obtained by data-dependent acquisition in both positive and negative ion modes.

An experienced analytical chemist with more than 10 years of experience in analyzing lipidomic data curated the annotation results of the MS-DIAL. The annotated spectra in the alignment files were manually evaluated. In total, 82,397 annotated spectra from 158 alignment result files derived from 78 positive and 80 negative ion mode data were curated. The annotation was labeled as “incorrect (false hit)” in the cases where the peak shape looked noisy (signal-to-noise ratio approximately less than 10) or the diagnostic ions were matched with barcode (noise) ions. This data curation process was not conducted using a systematic method; instead, the final judgment for “correct”, “mix”, and “false” labelings was made based on the knowledge and experience of the skilled technician. It is important to note that the labeling work was not performed for all annotated peaks. In our study, lipid quantification was performed using a representative adduct form that is determined for each lipid subclass independently [8]. For example, phosphatidylcholine (PC) is commonly detected in adduct ion forms such as [M + H]^+^, [M + Na]^+^, and [M + CH_3_COO]^−^, whereas only [M + CH_3_COO]^−^ is used to quantify lipid metabolites. Therefore, the annotated MS/MS spectra derived from PC’s [M + H]^+^ and [M + Na]^+^ are often not labeled in routine work. As a result, machine learning of the spectra not targeted by our group may be insufficient.

A total of 39,871 spectra encompassing 4100 metabolites of 116 lipid subclasses were labeled as “correct” (Appendix A, marked as “ok” in comment column). Among the spectral records available for training, we excluded certain lipid classes from the learning process. First, lipid classes with five or fewer records were excluded. This criterion was applied separately to both positive ion mode and negative ion mode data. The spectral records of free fatty acids (FA) and *N*-acyl ethanolamine (NAE) were also excluded because the annotations were performed only based on the information of RT and *m*/*z* values due to the lack of diagnostic criteria in the MS/MS spectra. Moreover, lipid spectral records having “others” as the ontology term were excluded: in the MS-DIAL lipidomics project, standard compound spectral library-based peak annotation based on the records of MassBank, GNPS, and NIST is also executed when the spectrum is not characterized by the rule-based lipid annotation pipeline. Finally, 17,029 spectral records of 3944 unique lipid molecules across 97 lipid classes were used for the machine learning (Appendix A). It contained 8451 ESI(+)-MS/MS spectra of 2559 unique lipids and 8578 ESI(-)-MS/MS spectra of 2048 unique lipids.

### 2.2. Spectral Data for the Validation of MS2Lipid

Annotated tandem mass spectral data from 31 projects were downloaded from the RIKEN LIPIDOMICS website (Appendix A). These spectral data were curated by different analysts from the training data used in this study. Lipidomic data were obtained using various MS machines, including a SCIEX TripleTOF 5600+ (index 18 to 28), Waters XevoG2 QTOF (index 78), ThermoFisher Q-Exactive Plus (index 79), Agilent 6546 QTOF system (index 80), SCIEX TripleTOF 6600 using SWATH-DIA (index 81), and Bruker timsTOF Pro (index 83 and 84). In addition, several studies (indexes 1 to 11) were obtained using the same analytical instrument (SCIEX TripleTOF 6600) but curated by different analysts.

### 2.3. Data Preprocessing for Machine Learning

Product ions (PLs) ranging from *m*/*z* 70–1250 were used, and the *m*/*z* values were rounded to one decimal place. The centroid peak heights in the bins were summed. In addition to the vector of the product ions, neutral loss (NL) information from the precursor *m*/*z* was used for the variables. The NL value was defined as the mass difference between the precursor *m*/*z* and product ion *m*/*z* values. The NL values ranging from 0 to 10 were excluded. The intensity of the NL vector was prepared in the same manner as that of the PL vector. The *m*/*z* and NL values with zero standard deviations in the dataset were excluded. Peak intensities were normalized, and the base peak intensity was standardized to 1 (Appendix A). In addition to the PL and NL vectors, a new descriptor, termed the mode of carbon and hydrogen (MCH) value, was created. The descriptor was designed to be the same within the same lipid subclass and differentiate lipid subclasses with similar spectral patterns. For example, the *m*/*z* values of the acetate adduct form of PC 33:1 (C_41_H_80_NO_8_P) and PC O-34:1 (C_42_H_84_NO_7_P) were 804.576 and 804.612, respectively, where the rounded precursor values were the same and the MS/MS spectral patterns were indistinguishable. Because the exact mass difference between PC 33:1 and PC O-34:1 arises from the exact masses of CH_4_ and O, the MCH value was defined by the following equation (Appendix A).
(1)$$mod\;by\;{CH}_{2}=m/z\;mod\;\left(exact\;mass\;of\;{CH}_{2}\right)$$
(2)$$mod\;by\;{CH}_{2}\;and\;H_{2}=\left(mod\;by\;{CH}_{2}\right)\;mod\;(exact\;mass\;of\;H_{2})$$
(3)$$factor=\left(exact\;mass\;of\;14H\right)\;mod\;\left(exact\;mass\;of\;{CH}_{2}\right)$$
(4)$$MCH=(mod\;by\;{CH}_{2}\;and\;H_{2})\;mod\;factor$$

The MCH value was designed to provide a consistent value among molecules of each lipid subclass (Appendix A) and was calculated based on the modulus operation to calculate the remainder in the division process. The remainder of the precursor *m*/*z* value based on the exact mass of CH_2_ was calculated (Equation (1)). Furthermore, the reminder value was dedicated to the modulus operation by the exact mass of H_2_ where the differences in acyl chain length and double bond number were canceled with the condition of fewer than six double bonds (Equation (2)). To provide a consistent value that was not affected by the double bond number, the correction factor was calculated (Equation (3)). A correction factor was applied to calculate the remainder generated using Equation (2) (Equation (4)). The MCH value yields the same value for molecules within the same lipid subclass, even if they have different fatty acid side chains. Unlike the Kendrick mass defect, which uses the precise mass difference from saturated hydrocarbons to estimate molecular formulas, the MCH value can be considered a descriptor specific to lipid molecules.

### 2.4. Machine Learning Model to Predict Lipid Subclass

Machine learning was performed using Python version 3.10.14. One-fifth (20%) of the dataset was used as the test set. The train_test_split function from scikit-learn “https://scikit-learn.org/stable/# (accessed on 1 April 2024)” was employed, with the stratify parameter set to the ontology data to ensure that the proportion of class labels remained consistent across both the training and test sets. In this study, we evaluated Support Vector Machines (SVM), k-Nearest Neighbors (KNN), Random Forest (RF), XGBoost (XGB), and Multi-layer Perceptron (MLP). XGBoost was implemented using the XGBoost package “https://xgboost.readthedocs.io/en/stable/index.html (accessed on 1 April 2024)”, while scikit-learn implementations were used for the other models. The performance of each model was assessed based on accuracy, recall, precision, and F1 score. For each method, parameters were optimized using GridSearchCV from scikit-learn with five-fold cross-validation, with accuracy serving as the selection criterion. The following parameters were optimized: SVM, regularization parameter and kernel function; KNN, number of neighbors; RF, number of estimators and maximum tree depth; XGB, tree depth, maximum tree depth, and learning rate; MLP, number of hidden layers and units, activation function, optimization algorithm, regularization parameter, and learning rate. As a result of the parameter optimizations, XGB demonstrated the best performance and was selected for further analysis. The MS2Lipid XGB model was constructed using ESI(+)-MS/MS and ESI(-)-MS/MS spectral records. The program was developed on a LINUX operating system having Intel(R) Xeon(R) Gold 6438M 64-Core Processor (max 2.2 GHz, min 0.8 GHz) with random access memory (RAM) of 1056 GB. In positive ion mode, the input and output feature sets comprised 1367 variables and 63 lipid subclasses, respectively. The optimized parameters were as follows: tree depth, 200; maximum tree depth, 3; and learning rate, 0.1. In negative ion mode, the input and output feature sets consisted of 1139 variables and 69 lipid subclasses, respectively, with the same parameter settings. The output layer represents the probability of the lipid ontology classification. The lipid subclass with the highest probability value was used as a representative candidate for the predicted lipid subclasses. In this study, if the representative lipid subclass was matched with the correct lipid subclass, the result became “correct”. If the correct lipid subclass was listed in the predicted lipid candidates where the probability value exceeds 1%, the result was defined as “listed in candidates”.

### 2.5. Using Shapley Additive exPlanations (SHAP) for Important Variables Interpretation

The important variables for XGBoost machine learners were evaluated using the SHAP values [15]. Using the SHAP function in Python 3.9-12 “https://github.com/shap/shap (accessed on 1 April 2024)”, 10 major lipid subclasses in each ion mode were assessed because of computational cost limitations. The evaluation was conducted using the lipid subclass’s spectral data in the test dataset.

### 2.6. Evaluation of CANOPUS for the Benchmark of MS2Lipid

The spectral records of the test datasets were analyzed using the CANOPUS ontology prediction program implemented in SIRIUS [16]. Ontology terms followed the definition of ClassyFire [2]. The records were converted to the “ms” format files as defined in the program. SIRIUS version 5.8.2 was utilized. To compare the results of MS2Lipid and CANOPUS, the subclass definition of LipidMAPS/MS-DIAL was converted to the definition of ClassyFire “class” definition (Appendix A). In this study, 13 ClassyFire classes were identified: fatty acids and conjugates, glycerophosphocholines, glycerophosphoethanolamines, glycerophosphoinositols, glycerophosphates, glycerophosphoserines, glycerophosphoglycerol, phosphosphingolipids, glycosphingolipids, ceramides, diacylglycerols, monoacylglycerols, and triacylglycerol.

### 2.7. Reanalysis of a Publicly Available Human Feces Lipidomics Data

Publicly available LC-ESI(+)-MS/MS- and LC-ESI(-)-MS/MS datasets from a human cohort study about the associations between insulin resistances and microbiome metabolisms in 306 individuals were downloaded from RIKEN DROPMet website (index DM0037). The LC-MS data were analyzed using MS-DIAL 5.1.6 [17] with the parameter set available in Appendix A, where the retention time correction function was used to improve the peak alignment process (Appendix A). The alignment results for the positive and negative ion modes are presented in Appendix A. For the spectrum network analysis, the MS/MS spectral similarity was calculated using the Bonanza scoring system of MS-DIAL [18] (see Appendix A for the parameter details), and was visualized using Cytoscape version 3.10.1 “https://cytoscape.org/ (accessed on 1 April 2024)”. For the correlation analysis between the lipidome and microbiome data, 26 bacterial genera with less than 50% of samples having zero counts were used. The relationship between lipids and the gut microbiota was evaluated using Spearman’s correlation. 

## 3. Results

### 3.1. Vector Construction and Descriptor Development for Lipid Subclass Differentiation in Tandem Mass Spectra

We utilized 82 studies comprising a total of 82,397 spectra for vector construction. The MS-DIAL program [9] was employed for peak picking, annotation, and peak alignment. After the curation for annotated peaks (see Section 2), our study employed a total of 17,029 spectra from 3944 unique lipids belonging to 97 subclasses. This included 8451 ESI(+)-MS/MS spectra of 2559 unique lipids and 8578 ESI(-)-MS/MS spectra of 2048 unique lipids. The mass spectrum is represented as a vector through the following procedure (Figure 1) (also see Section 2). The high-resolution mass values were converted to nominal mass values using a simple rounding method. In addition to the vector of product ions, the neutral loss (NL) from the precursor *m*/*z* value was also included as a variable. Furthermore, an additional descriptor was created for this study. The descriptor is designed to distinguish between two independent subclasses that have very similar spectra but can be distinguished by their high-resolution *m*/*z* values (Appendix A). An exemplary case is the differentiation between PC and ether-linked PC. For instance, PC 15:0_18:1 and PC O-16:0_18:1 have molecular formulas and exact mass values of C_41_H_80_NO_8_P and 745.562, and C_42_H_84_NO_7_P and 745.599, respectively, resulting in a 37 mDa mass difference. These metabolites can be distinguished using high-resolution MS. Therefore, we created a value called “mod by carbon-hydrogen mass values (MCH-value)”, which provides a consistent value for each lipid subclass (Appendix A).

### 3.2. Selection and Optimization of Machine Learning Models

We evaluated five machine learning methods: k-nearest neighbor (KNN), random forest (RF), support vector machine (SVM), XGBoost (XGB), and Multi-layer Perceptron (MLP) (Figure 2a). The results indicated that the model using XGBoost offered the best accuracy in both ion modes. We evaluated the optimized model using the test dataset, in which the lipid subclass with the highest probability value was defined as the representative candidate. In the case that the correct lipid subclass was predicted by the probability of >1%, the result was defined as “listed in candidates” (Figure 2b). The result showed that the correct lipid subclass was predicted with accuracies of 97.2% and 97.6% for the positive and negative ion mode spectra, respectively. Moreover, the correct lipid subclass was listed as a candidate for more than 99% of the queries. In addition, an investigation was conducted to ascertain how the prediction accuracy is affected when the lipid class labels are randomly shuffled. Models constructed using data with shuffled labels exhibited an accuracy of 1.7% for positive ion mode and 1.6% for negative ion mode. The decrease of prediction accuracy suggests that the model is not merely modelling inherent noise but is effectively learning lipid class-specific spectral patterns. Consequently, these findings indicate that MS2Lipid was adequately trained and generated meaningful classifications based on the data.

The Shapley additive explanation (SHAP) score was calculated to investigate the important descriptors for lipid subclass classification (Figure 2c) [14]. Owing to the limitations of the SHAP calculation cost, the important features for predicting the 10 lipid subclasses were evaluated for each ion mode. First, the “MCH-value” descriptor was extracted as the most important feature regardless of ion mode. In addition, the product ions of *m*/*z* 184 and *m*/*z* 369 and the neutral loss (NL) of 141 Da were extracted as the top three important features in the positive ion mode, which denote the phosphocholine polar head (C_5_H1_5_NO_4_P^+^, *m*/*z* 184.073), cholesterol aglycone (C_27_H_45_^+^, *m*/*z* 369.349), and phosphoethanolamine polar head (C_2_H_8_NO_4_P, NL of 141.019 Da), respectively. In addition, in the negative ion mode, NL of 74, NL of 197 Da, and *m*/*z* 153 were extracted as the top three important features, which denote the loss of acetic acid plus methyl (C_3_H_6_O_2_, 74.037 Da) in EtherPC and PC, the loss of ethanolamine glycerophosphate (C_5_H_14_NO_5_P, 199.061 Da) observed in lyso PE (LPE), and the product ion of glycerol phosphate (C_3_H_6_O_5_P^−^, *m*/*z* 152.987) observed in glycerophospholipids, respectively. These SHAP results indicate that the MS2Lipid model recognizes diagnostic ions that have been utilized in the lipidomics community to predict lipid subclasses. The accuracy of MS2Lipid for the test set was 97.4% (Appendix A).

### 3.3. The Performance Comparisons Between MS2Lipid and CANOPUS

We used CANOPUS [12] as the benchmark for the MS2Lipid program, which classifies product ion spectra into chemical classes defined by ClassyFire. The same test set was used for both programs. It is important to note that CANOPUS predicts all chemical classes, including lipids, from the query spectrum. The model was trained using authentic standard-derived spectral libraries such as GNPS and MassBank. However, the spectral records of lipids in these libraries are smaller compared to our training set. Therefore, the following comparison reflects the lipid diversity in the training sets rather than the model-building process. For example, our training dataset includes 5224 spectra from 1408 molecules of glycerophosphocholines, glycerophosphoethanolamines, glycophosphoinositols, glycerophosphates, glycerophosphoserines, and glycerophosphoglycerols. In contrast, the GNPS (GNPS-LIBRARY as of 25 January 2024) and MassBank (version 2023.11) databases have 60 spectra from 43 molecules and 2579 spectra from 558 molecules, respectively [19,20]. Additionally, our training set includes 1301 spectra from 456 triacylglycerols (TGs), while the GNPS and MassBank databases have seven spectra from five molecules and sixty-five spectra from seventeen molecules, respectively (Appendix A). Since our model generates lipid ontologies defined by LIPID MAPS or MS-DIAL, the output of the MS2Lipid program was converted to the compound class defined by ClassyFire. The accuracy of the MS2Lipid program in predicting the ClassyFire “class” level exceeded 99.0%. CANOPUS did not generate output for half of the spectral queries because it provides predicted results only for compounds with a probability above 50% and primarily supports computation for compounds less than 600 Da “https://boecker-lab.github.io/docs.sirius.github.io/ (accessed on 1 April 2024)”.

CANOPUS supported only 38.7% of the spectral queries. However, the MS2Lipid program outperformed CANOPUS in prediction accuracy (Appendix A), with CANOPUS achieving 82.4% accuracy for the queries that CANOPUS could handle (Figure 3). This result indicates that in lipid spectrum machine learning research, it is necessary to accumulate both standard spectra and spectral information from biological samples to build a highly accurate learning model.

### 3.4. Evaluation of Robustness and Scalability of MS2Lipid by Using Various Spectral Data

We evaluated the robustness of MS2Lipid for spectra obtained using various machines and curated by various analysts. The data resources were downloaded from the RIKEN LIPIDOMICS website, in which the spectral data from Waters, Bruker, Thermo Fisher, Agilent, and SCIEX were available. The spectral data in our training dataset were not included in the data resource. First, the MS/MS spectral data acquired by the same machine and curated by different analysts were imported into the MS2Lipid program, resulting in 89.7% and 84.5% accuracy for the positive and negative ion modes, respectively (Figure 4a). This is an important result because different data curators may offer different results, which reminds us of developing a machine learning model that provides an orthogonal decision objectively, regardless of the data analysts. Next, we applied the MS2Lipid program to the spectra acquired using different curators and MS techniques (Figure 4b). While the accuracies ranged from 77% to 98%, the accuracies for the negative ion mode of Waters, the negative ion mode of Agilent, and the positive ion mode of Bruker were relatively low. According to the SHAP values (Figure 2c), the MS2Lipid machine learning model constructed in this study positions the MCH value as the most important variable, which suggests that the results are highly influenced by the accuracy of the precursor *m*/*z*. Therefore, we investigated the distribution of measurement errors between the observed and theoretical values of the lipid *m*/*z* included in the RIKEN lipidomics data (Appendix A). The result indicated that the measurement errors for the negative ion mode of Waters and Agilent and the positive and negative ion modes of Bruker and SCIEX were larger compared to the results of the positive and negative ion modes of Thermo Fisher and the positive ion mode of Agilent, suggesting that the MS2Lipid model may place importance not only on the patterns of the product ion spectrum but also on the accuracy of the precursor *m*/*z*.

Finally, we demonstrated how the MS2Lipid program handled the product ion spectra derived from co-eluted lipids (Appendix A). An MS/MS spectrum considered to be a mixture of PC and PE molecules because of the presence of *m*/*z* 184 for the PC motif and a neutral loss (NL) of 141 for the PE motif is shown. The MS2Lipid program provided probabilities as the results, in which the mixed spectra were predicted as “95.7% PE and 2.2% PC”, while the MS-DIAL program only offers either PC or PE as a representative candidate based on the similarity score; the percentages do not represent the abundance ratio of the two metabolites. Even in cases where manual inspection of the mixed spectra of PC and PE suggests that both PE and PC are present to a similar extent, there was a tendency for a high probability to be assigned to either PE or PC. However, when the criterion adopted in this study, which considers spectra with a presence probability of 1% or more to be ‘listed in candidates’, was applied to the mixed spectra of PE and PC, MS2Lipid suggested the presence of both PC and PE for all spectral datasets labelled as a mixture of PC and PE (Appendix A). These results suggest that MS2Lipid can assist in the interpretation of mixed spectra data.

### 3.5. Application of MS2Lipid for Annotating Lipids That Have Previously Been Unknown in a Human Cohort Analyzing the Fecal Lipidome

The MS2Lipid program was used to annotate the novel lipid molecules. We reanalyzed a publicly available LC-ESI(+)-MS/MS- and LC-ESI(-)-MS/MS datasets from a human cohort study where the associations between insulin resistances and microbiome metabolisms in 306 individuals were investigated [12]. The MS-DIAL program generated 22,980 and 12,742 unique peak features in ESI(+)- and ESI(-) datasets, respectively, in which the MS/MS spectra for 18,497 and 10,224 features (precursor ions) were assigned. Of these, 4135 of ESI(+) and 2967 of ESI(-) spectra were characterized by the MS-DIAL rule-based annotation algorithms in combination with the predicted retention time information.

First, we applied the MS2Lipid program to the annotated MS/MS spectra using MS-DIAL. As a result, 65.4% and 69.9% of peaks (if considered listed as candidates; 85.4% and 90.2%) for ESI(+)- and ESI(-)-MS/MS spectra, respectively, were predicted with the same lipid ontology, where the lipid subclasses supported by the current MS2Lipid program were evaluated. This indicated that one-third of the MS/MS spectra characterized by MS-DIAL were not assigned by the MS2Lipid program. In contrast to MS-DIAL, which estimates lipid structures based on the presence or absence of diagnostic ions without considering the effects of noise or contaminated ions, MS2Lipid determines lipid classes by considering the presence of noise and contaminated ions, reflecting the discretion of experienced analysts (Figure 2a). This suggests that MS2Lipid performs more conservative annotations, which could explain why one-third of the spectra by MS-DIAL were not supported by MS2lipid. Next, we applied the MS2Lipid program to the unknown spectra, in which 14,272 peaks with MS/MS spectral information were labeled as unknown in MS-DIAL. The molecular spectrum network was constructed (Figure 5a). The top-hit annotations by MS-DIAL and MS2Lipid were mapped onto the network, and the results were reflected by node fill and border colors, respectively. Distinct clusters based on major lipid subclasses were created in the molecular spectrum network. One cluster contains a subclass of bile acid esters assigned as steryl ester (SE) 24:1; O4/FA, whose well-known structural backbone is deoxycholic acid (DCA) and its isoform (isoDCA) is the second bile acid biosynthesized by bacteria [19]. Hereafter, we define the bile acid ester assigned by SE 24:1; O4/FA as “DCAE”, implying deoxycholic acid ester. Interestingly, unknown peak features were predicted to be DCAE and *N*-acyl glycine, and the unknown nodes were connected to those of DCAEs. The existence of the term *N*-acyl glycine in addition to DCAE enabled us to elucidate unknown molecules such as glycine-conjugated deoxycholic acid ester (GDCAE), which contains *m*/*z* 414 as a unique ion (Figure 5b). Because the stereochemistry of the sterol backbone cannot be characterized by the CID-MS/MS technique, the characterization of the molecules with *m*/*z* 414 was assigned as SE 24:1; O4; G/FA in our study, where the nomenclature follows the definition of LIPID MAPS. It has been reported that the bile acid of isoDCA is efficiently converted to its ester form [20]. Furthermore, neither SE 24:1; O4; G/FA nor SE 24:1; O4/FA were detected in our previous study analyzing mouse feces [21], and isoDCA was less abundant in mouse feces than in human feces [18]. Furthermore, the ester form of ST 24:1; O5, implying cholic acid, was not detected in human feces, which was also true in mouse feces. This indirect evidence enabled us to expect the structure to be a glycine-conjugated isodeoxycholic acid ester, although its complete structure should be confirmed using an authentic standard in the future.

We explored the relationship between the amount of GDCAE molecules and the clinical diagnosis of obese and healthy individuals (Figure 5c). The abundance of glycine-conjugated molecules, including FA 16:0, 18:1, 18:2, 20:4, and 20:5, was significantly increased in obese individuals. Furthermore, correlation analysis between bile acids and bacteria revealed that the abundance of DCAE was highly correlated with *Alistipes*, *Eubacterium*, *Faecalibacterium*, and *Subdoligranulum*, whereas the abundance of GDCAE molecules was negatively correlated with *Alistipes*, *Blautia*, and *Prevotella*. Given that the highly abundant molecules of DCAE and GDCAE contained FA 16:0 (palmitic acid), 18:1 (putatively oleic acid), and 18:2 (putatively linoleic acid), the ratios of DCAE and GDCAE clearly correlated with the abundance of *Alistipes*. Since *Alistipes* and its phylum *Bacteroidota* are known to have bile salt hydrolases (BSH) enzymatic activity, the ratio of conjugated to deconjugated bile acid esters may be attributable to *Bacteroidota*, including *Alistipes* [22]. Likewise, we characterized additional three conjugated bile acid candidates, which included SE 24:1; O3; G/FA (glycine conjugated lithocholic acid ester, GLCAE, as a representative structure), SE 24:1; O4; T/FA (taurine conjugated deoxycholic acid ester, TDCAE, as a representative structure), and SE 24:1; O3; T/FA (taurine conjugated lithocholic acid ester, TLCAE, as a representative structure). These bile acid esters were validated by checking the diagnostic ions to recognize the aglycone and free fatty acid features (Appendix A). Furthermore, mannosylinositol phosphorylceramide (MIPC) was also characterized in the human cohort data because of the existence of phosphatidylinositol ceramide (PI-Cer) classification by the MS2lipid program, in addition to a unique neutral loss of hexose [23]. It is known that not only bacteria but also fungi exist in the human gut [24]. The detection of MIPC, a lipid produced by fungi, in a human cohort study was reasonable.

## 4. Discussion

The MS2Lipid program has been developed to provide probability scores for lipid subclasses in MS/MS queries using a machine learning approach. It has been trained on over 10,000 lipid MS/MS spectral data and has shown accurate lipid subclass estimation with 97.4% accuracy. Compared to the CANOPUS metabolite ontology prediction program, MS2Lipid outperformed in terms of performance. Unlike MS-DIAL, which provides the most probable lipid candidate for a single query, MS2Lipid presents estimated lipid subclasses along with their probabilities. This feature is particularly useful for understanding mixed spectra of multiple co-eluting metabolites. Additionally, we demonstrated the potential application of MS2Lipid for structural estimation of unknown spectra when used in combination with molecular spectrum networking.

The MS-DIAL program introduced in 2020 offers accurate and comprehensive lipid annotation using rule-based fragment ion annotations with predicted retention time information. However, the annotation results are often revised through manual curation by experienced technicians. The reasons why data curators modify the MS-DIAL results vary. For example, while fatty acids such as 18:4 or 18:5 typically do not exist in mammalian cells, complex lipids containing them are often suggested in the MS-DIAL software program. Such annotations of minor fatty acids may be true; however, conservative analysts often exclude these annotations. Moreover, even if a spectrum can be clearly interpreted, annotations are omitted if the chromatographic peak shape is poor or if the peak has a low signal-to-noise ratio, which makes quantitative discussion difficult. Similarly, when a product ion spectrum can be interpreted as a mixture of PC and PE, as demonstrated in this study, the peak feature is often excluded from the lipidome table in statistical analyses because the quantitative discussion becomes challenging. From 2020 to 2023, our research group has been labeling each lipid annotation result output by MS-DIAL with “confidence (correct)”, “mixed spectrum”, “uncertainty”, and “misannotation (false hits)” labels, by using MS-DIAL’s graphical user interface (GUI). The training set used in this study only utilized those labeled with “confidence (correct)”, reflecting high certainty, and moreover, it reflects the labeling results of just a single experienced analyst.

Machine learning from training sets derived from a single curator has both advantages and disadvantages. The advantage is the ability to maintain data quality. Additionally, it provides clear decision-making for spectral queries, although judgments may vary among individuals. In untargeted lipidomics, which is oriented towards discovery science, some researchers opt to curate spectra in a way that tolerates a certain level of false positives, aiming to reduce false negatives. Stearidonic acid, an omega-3 fatty acid derived from plant oils, is a representative candidate for 18:4 mentioned above, and lipids containing 18:4 can be detected in the human body [25]. The possible intake of such exposomes introduces ambiguity, leading to variations in data curation among analysts. In fact, our results in the MS2Lipid evaluation showed that differences among data curators had a greater impact on accuracy estimation than machine differences. Thus, decisions that reflect the will of an analyst can be interpreted as both strengths and weaknesses. This disadvantage may be due to the slow pace of spectral data accumulation. However, from the perspective of maintaining data quality, this seems to be a tradeoff. We plan to continue developing our approach, including quality control methods, as we accumulate more spectra for machine learning in the future. Furthermore, the study investigated a newly characterized lipid subclass, GDCAE, in relation to microbiome data. GDCAE molecules were found to be significantly increased in obese individuals (Figure 5c), which is associated with the decreased activity of BSH, a factor linked to obese-related diseases such as type-2 diabetes [26]. The abundances of DCAE and GDCAE were highly correlated with the abundances of the *Alistipes* genus, which has shown improvements in lipid accumulation and insulin resistance through the reduction of gut monosaccharide levels [12]. Understanding the impact of glycine-conjugated bile acid esters on human intestinal immune mechanisms and their relationship with bacteria, including *Alistipes*, may provide valuable insights for probiotics and prebiotics. MS2Lipid is not only a tool for lipid annotation but also a platform for gaining new insights into lipid biology. Future improvements to MS2Lipid aim to predict not only lipid subclasses but also more detailed structures, including fatty acid acyl chains, to contribute to the advancement of untargeted lipidomics.

## 5. Conclusions

We constructed the MS2Lipid model by using manually curated MS/MS spectra for predicting lipid subclasses in untargeted lipidomics. The model achieved a high classification accuracy of 97.4% across 97 lipid subclasses. Furthermore, additional validation using various datasets from different instruments and curators yielded an average accuracy exceeding 87.2%. The accuracy can be improved by incorporating the information of retention times of lipid molecules. Moreover, we demonstrated the utility of MS2Lipid through an integrated approach with molecular spectral networking by annotating microbiota-derived esterified bile acids, whose abundances were significantly increased in fecal samples from obese patients. This suggests that the machine learning model provides an independent and orthogonal criterion for lipid subclass classification and has the potential to accelerate annotation of previously unknown lipids with cheminformatics approaches such as molecular spectrum networking. In future, the program will be updated to annotate fatty acid composition and *sn*-positional information, and to reduce false positive and negative annotations in the results of lipidomics software programs. In addition, a computational approach should be developed to predict novel lipid structures that are not yet known, as there are still many unknown lipids among the metabolites produced by bacteria and fungi present in the microbiome.

## Figures and Tables

**Figure 1 metabolites-14-00602-f001:**
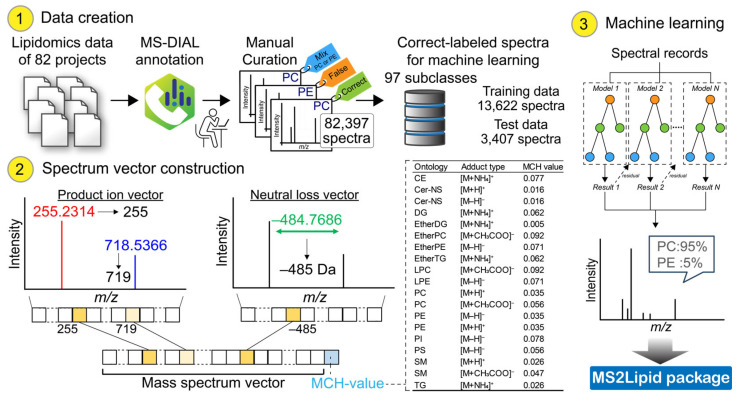
Workflow to create MS2Lipid machine learning model. In the data creation step, the spectral data were derived from 82 projects. The lipidomics data were analyzed by MS-DIAL, providing lipid annotation for 82,397 spectra that include positive and negative ion modes. The original annotation was manually curated, and labeled as correct, mix, or false hit. The correctly labeled data of 17,029 spectra were divided into training and test sets. In the spectrum vector construction step, the MS/MS spectrum is represented by the array of product ions and neutral losses whose high-resolution mass value is converted into nominal mass. In addition, a descriptor “MCH-value” was calculated from the accurate precursor *m*/*z* value. The descriptors for cholesteryl ester (CE), ceramide containing sphingosine and normal fatty acid (Cer-NS), diacylglycerol (DG), ether-linked DG (EtherDG), phosphatidylcholine (PC), lysoPC (LPC), ether-linked PC (EtherPC), triacylglycerol (TG), ether-linked TG (etherTG), phosphatidylethanolamine (PE), lysoPE (LPE), phosphatidylinositol (PI), phosphatidylserine (PS), and sphingomyelin (SM) were described for the major adduct types. The vectors were used as the input for machine learning models. The output shows the probability ratio of lipid subclass classifications.

**Figure 2 metabolites-14-00602-f002:**
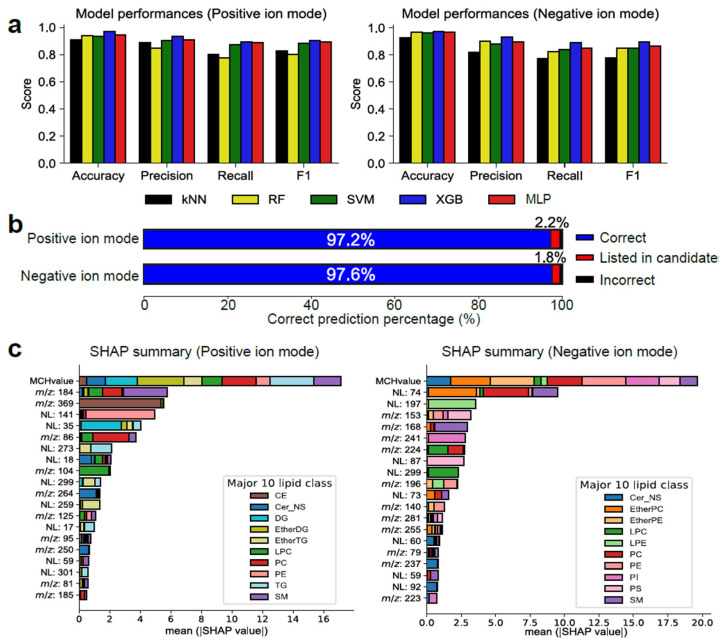
Evaluation of machine learning models. (**a**) Accuracy, precision, recall, and F1-score in each model, where k-nearest neighbor (KNN), random forest (RF), support vector machine (SVM), XGBoost (XGB) and Multi-layer Perceptron (MLP) were evaluated. (**b**) Prediction accuracy in the XGBoost model. The upper and lower panels correspond to the results in positive and negative ion modes, respectively. The output becomes correct if the predicted lipid subclass is equal to the correct label, and the result becomes “listed in candidates” if the correct label exists in the candidate list. (**c**) The Shapley additive explanation (SHAP) scores to investigate important descriptors. The top 20 SHAP features for predicting 10 major lipid subclasses were described. NL, neutral loss; phosphatidylethanolamine; PI, phosphatidyl-inositol; PS, phosphatidylserine; EtherPE, ether-linked PE; LPC, lysoPC; LPE, lysoPE.

**Figure 3 metabolites-14-00602-f003:**
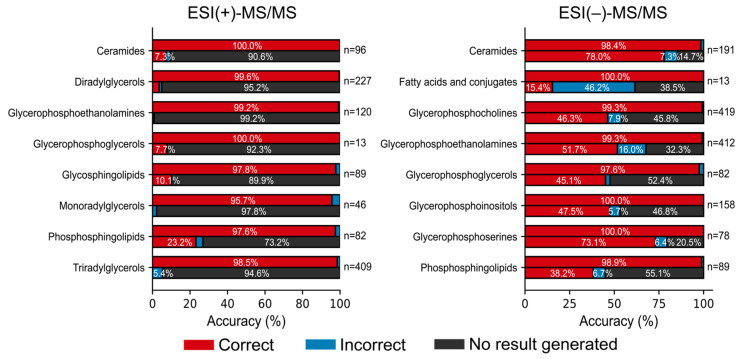
Comparison of MS2Lipid with CANOPUS. The left and right panels show the results of positive and negative ion mode, respectively. In each lipid class, the upper and lower panels are the results of MS2Lipid and CANOPUS, respectively. The accuracy (%) of ClassyFire class level ontology prediction was described. The number of molecules in each ontology was also shown.

**Figure 4 metabolites-14-00602-f004:**
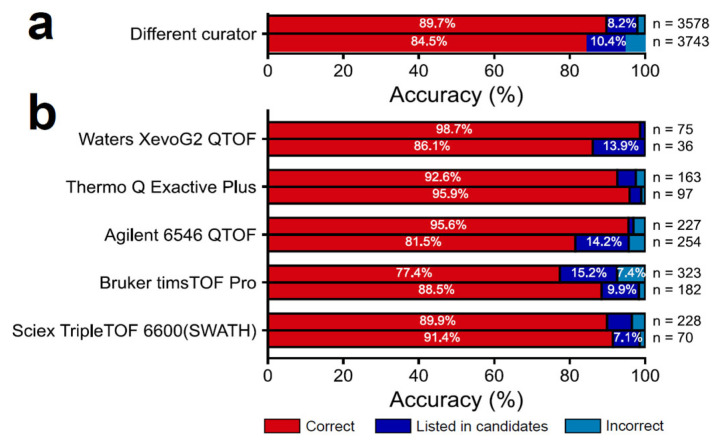
Evaluation of robustness of MS2Lipid. (**a**) Validation using the spectral data by different curators with the completely same MS machine. (**b**) Validation using the spectral records obtained by different instruments and different curators. The outputs are categorized into three cases: if the generated label of MS2Lipid is the same as the query’s label, the result becomes “correct”; while the labels are not matched, the result becomes “listed in candidates” if the correct name is listed in the candidates predicted with more than 1% probability; and for the others, the result becomes “incorrect”.

**Figure 5 metabolites-14-00602-f005:**
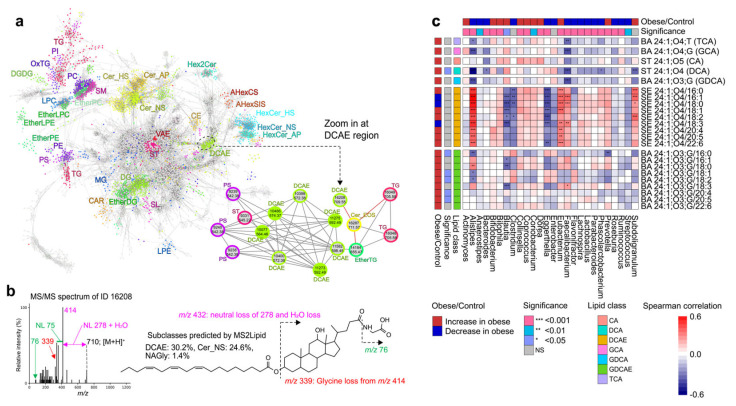
Elucidation of microbiota-derived lipid metabolites. (**a**) Molecular spectrum networking from peak features of positive ion mode in a human cohort study analyzing fecal samples. The node fill and border colors represent the annotation of MS-DIAL and MS2Lipid, respectively. (**b**) The tandem mass spectrum of newly characterized bile acid ester and its plausible structure were described. The features of *m*/*z* 76 and neutral loss (NL) of 75 originate from the glycine structure. The features of *m*/*z* 339, NL 296, and *m*/*z* 414 are derived from the lipid subclasses of deoxycholic acid, FA 18:3-containing lipid, and the glycolic acid structure, respectively. (**c**) Correlation analysis between bacteria and bile acid metabolites. The spearman correlation between bacteria and lipid abundances was calculated, where the statistical significance was estimated by t-test (two-sided). The significance of obese and control individuals was calculated by Mann–Whitney U test (two-sided). * *p* < 0.05, ** *p* < 0.01, and *** *p* < 0.001. CA; cholic acid, DCA; deoxycholic acid, DCAE; esterified deoxycholic acid, GCA; glycocholic acid, GDCA; glycodeoxycholic acid, GDCAE; esterified glycodeoxycholic acid, TCA; taurocholic acid.

## Data Availability

The training, validation, and test data for constructing the MS2lipid program are presented in Appendix A. The experimental MS/MS spectra for validation of MS2lipid were downloaded from the RIKEN Lipidomics website (http://prime.psc.riken.jp/menta.cgi/lipidomics/index, accessed on 1 April 2024). Human cohort lipidomics data were downloaded from the RIKEN DROPMet website (http://prime.psc.riken.jp/menta.cgi/prime/drop_index, accessed on 1 April 2024) under the index number DM0037. The source code and tutorial for MS2lipid are available at https://github.com/systemsomicslab/ms2lipid/tree/main/notebooks, (accessed on 1 April 2024).

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
