# Peer review of "MS2Lipid: A Lipid Subclass Prediction Program Using Machine Learning and Curated Tandem Mass Spectral Data"

_metabolites, 2024, doi:10.3390/metabo14110602_

Round 1
Reviewer 1 Report
Comments and Suggestions for Authors
The manuscript on lipid subclass prediction program using machine learning and curated tandem mass spectral data is very interesting, unique and well defined. The method used here are accurate and results well defined. I don’t have many suggestions in the manuscript as with best of my knowledge I don’t find any loopholes in any of the data. Although, considering technical points I would like to mention below points-
1. Authors can add few more keywords for the better visibility of the article.
2. I wonder, why conclusion is not added to the manuscript. I would suggest to add conclusion section featuring the future prospects of the current work.
3. Page 2, line 48, support your statement by latest research- 10.1016/j.jlr.2024.100600.
4. Make the references uniform.
Author Response
Comments 1: Authors can add few more keywords for the better visibility of the article.
Response 1: Thank you. We added three more keywords: lipid class prediction, microbiota-dependent lipids, and human fecal samples.
Comments 2: I wonder, why conclusion is not added to the manuscript. I would suggest to add conclusion section featuring the future prospects of the current work.
Response 2: We added conclusion section in our manuscript, where the future prospects of this work were described.
Comments 3: Page 2, line 48, support your statement by latest research- 10.1016/j.jlr.2024.100600.
Response 3: Thank you. We believe that the sentence of page 2 and line 48 indicates “Untargeted lipidomics, utilizing liquid chromatography coupled with tandem mass spectrometry (LC-MS/MS) and electrospray ionization (ESI), efficiently provides the profile of 500 to 1000 lipid molecules from a single biospecimen”. Since the paper that you suggested does not contain a lipidome table as a supplementary file to count the number of annotated lipids, we added a paper describing the number of annotated lipids in several mouse organs (doi:10.1038/s43587-024-00610-6.).
Comments 4: Make the references uniform.
Response 4: Thank you for pointing this out. We managed the citations in EndNote, and the references should be now uniformed.
Reviewer 2 Report
Comments and Suggestions for Authors
The work under review presents MS2Lipid, a novel program for lipid subclass prediction, which exploits machine learning techniques i.e. machine learning and tandem mass spectral (MS/MS) data. This study fits into a growing context of lipid metabolism research, where accurate annotation of lipid species plays a fundamental role in understanding their biological functions and their role in metabolic diseases. The software is based on a set of curated MS/MS spectra, which include a rich annotation of lipid subclasses. The curated dataset is used to train a machine learning model, capable of recognizing key patterns in the fragmentation spectrum of each lipid subclass. The central idea is that the lipid fragmentation spectrum contains sufficient information to distinguish not only between the main lipid classes, but also between the various subclasses, which often share similar structural features and are difficult to distinguish with traditional methods.
The work is certainly very interesting, however a notable issue of the work is represented by the iThenticate report (over 70% of copied text). The authors must necessarily modify the text in order to reduce this percentage.
One of the most interesting aspects of this study is the potential impact that MS2Lipid could have in biomedical research and metabolomic analysis. The ability to accurately predict lipid subclasses could accelerate the discovery of new lipid biomarkers and provide new information on lipid interactions in physiological and pathological processes. Furthermore, the software could be useful in clinical settings, for example for the identification of lipid profile alterations in diseases such as diabetes, obesity and cardiovascular disorders.
However, the study is not free from limitations that should be discussed. The main limitation concerns the dependence on curated data, which could restrict the ability of the model to be generalized to less accurately annotated datasets. In the future, it would be interesting to explore strategies to make the model more adaptable to heterogeneous datasets, as well as to extend its predictive capabilities to include even more specific lipid classes or those not covered by the datasets used. Furthermore, another possible area of ​​improvement concerns the scalability of the approach. The analysis of large volumes of MS/MS data may require optimizations in terms of computational efficiency, especially in resource-limited laboratory settings.
Author Response
Thank you very much for taking the time to review this manuscript. Please find the detailed responses below, and the major changes in the main text were highlighted in yellow for your ease of review.
Comments 1: The work is certainly very interesting, however a notable issue of the work is represented by the iThenticate report (over 70% of copied text). The authors must necessarily modify the text in order to reduce this percentage.
Response 1: Thank you for checking our manuscript with iThenticate. We have also used the iThenticate program to reproduce your result, which shows a total similarity index of 76%. However, 61% of this matches the bioRxiv preprint paper of our manuscript, and the remaining sentences were only matched with the references in the References section. Therefore, we believe that our manuscript is suitable for a scientific paper.
Comments 2: In the future, it would be interesting to explore strategies to make the model more adaptable to heterogeneous datasets, as well as to extend its predictive capabilities to include even more specific lipid classes or those not covered by the datasets used.
Response 2: Thank you very much. In this revision, we added a conclusion section that includes several future perspectives, including the prediction of novel lipid structures.
Comment 3: Furthermore, another possible area of improvement concerns the scalability of the approach. The analysis of large volumes of MS/MS data may require optimizations in terms of computational efficiency, especially in resource-limited laboratory settings.
Response 3: Thank you so much for your constructive comments. Since the MS2Lipid model is not complicated, we did not worry about the optimizations in terms of computational efficiency. To clarify the computational environment used in this study, we added the computing environment information for the MS2Lipid program using XGBoost.
Round 2
Reviewer 2 Report
Comments and Suggestions for Authors
The authors have addressed the comments.